# A Minimax Approach to Supervised Learning

**Farzan Farnia**[*]
farnia@stanford.edu

**David Tse**[*]
dntse@stanford.edu

## Abstract

Given a task of predicting $Y$ from $X$, a loss function $L$, and a set of probability distributions $\Gamma$ on $(X, Y)$, what is the optimal decision rule minimizing the worst-case expected loss over $\Gamma$? In this paper, we address this question by introducing a generalization of the maximum entropy principle. Applying this principle to sets of distributions with marginal on $X$ constrained to be the empirical marginal, we provide a minimax interpretation of the maximum likelihood problem over generalized linear models as well as some popular regularization schemes. For quadratic and logarithmic loss functions we revisit well-known linear and logistic regression models. Moreover, for the 0-1 loss we derive a classifier which we call the minimax SVM. The minimax SVM minimizes the worst-case expected 0-1 loss over the proposed $\Gamma$ by solving a tractable optimization problem. We perform several numerical experiments to show the power of the minimax SVM in outperforming the SVM.

## 1 Introduction

Supervised learning, the task of inferring a function that predicts a target $Y$ from a feature vector $\mathbf{X} = (X_1, \ldots, X_d)$ by using $n$ labeled training samples $\{(\mathbf{x}_1, y_1), \ldots, (\mathbf{x}_n, y_n)\}$, has been a problem of central interest in machine learning. Given the underlying distribution $\tilde{P}_{\mathbf{X}, Y}$, the optimal prediction rules had long been studied and formulated in the statistics literature. However, the advent of high-dimensional problems raised this important question: What would be a good prediction rule when we do not have enough samples to estimate the underlying distribution?

To understand the difficulty of learning in high-dimensional settings, consider a genome-based classification task where we seek to predict a binary trait of interest $Y$ from an observation of $3,000,000$ SNPs, each of which can be considered as a discrete variable $X_i \in \{0, 1, 2\}$. Hence, to estimate the underlying distribution we need $O(3^{3,000,000})$ samples.

With no possibility of estimating the underlying $\tilde{P}$ in such problems, several approaches have been proposed to deal with high-dimensional settings. The standard approach in statistical learning theory is empirical risk minimization (ERM) [1]. ERM learns the prediction rule by minimizing an approximated loss under the empirical distribution of samples. However, to avoid overfitting, ERM restricts the set of allowable decision rules to a class of functions with limited complexity measured through its VC-dimension.

This paper focuses on a complementary approach to ERM where one can learn the prediction rule through minimizing a decision rule's worst-case loss over a larger set of distributions $\Gamma(\hat{P})$ centered at the empirical distribution $\hat{P}$. In other words, instead of restricting the class of decision rules, we consider and evaluate all possible decision rules, but based on a more stringent criterion that they will have to perform well over all distributions in $\Gamma(\hat{P})$. As seen in Figure 1, this minimax approach can be broken into three main steps:

1. We compute the empirical distribution $\hat{P}$ from the data,

---

[*]Department of Electrical Engineering, Stanford University, Stanford, CA 94305.

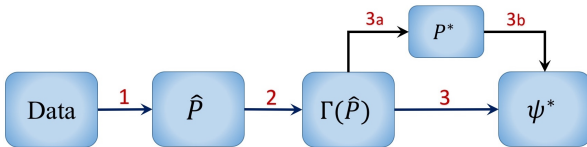

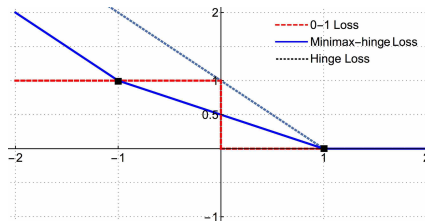

Figure 1: Minimax Approach                    Figure 2: Minimax-hinge Loss

2. We form a distribution set $\Gamma(\hat{P})$ based on $\hat{P}$,

3. We learn a prediction rule $\psi^*$ that minimizes the worst-case expected loss over $\Gamma(\hat{P})$.

Some special cases of this minimax approach, which are based on learning a prediction rule from low-order marginal/moments, have been addressed in the literature: [2] solves a robust minimax classification problem for continuous settings with fixed first and second-order moments; [3] develops a classification approach by minimizing the worst-case hinge loss subject to fixed low-order marginals; [4] fits a model minimizing the maximal correlation under fixed pairwise marginals to design a robust classification scheme. In this paper, we develop a general minimax approach for supervised learning problems with arbitrary loss functions.

To formulate Step 3 in Figure 1, given a general loss function $L$ and set of distribution $\Gamma(\hat{P})$ we generalize the problem formulation discussed at [3] to

$$\operatorname*{argmin}_{\psi \in \mathbf{\Psi}} \max_{P \in \Gamma(\hat{P})} \mathbb{E}\left[ L\big(Y, \psi(\mathbf{X})\big) \right]. \tag{1}$$

Here, $\mathbf{\Psi}$ is the space of all decision rules. Notice the difference with the ERM problem where $\mathbf{\Psi}$ was restricted to smaller function classes while $\Gamma(\hat{P}) = \{\hat{P}\}$.

If we have to predict $Y$ with no access to $\mathbf{X}$, (1) will reduce to the formulation studied at [5]. There, the authors propose to use the *principle of maximum entropy* [6], for a generalized definition of entropy, to find the optimal prediction rule minimizing the worst-case expected loss. By the principle of maximum entropy, we should predict based on a distribution in $\Gamma(\hat{P})$ that maximizes the entropy function.

How can we use the principle of maximum entropy to solve (1) when we observe $\mathbf{X}$ as well? A natural idea is to apply the maximum entropy principle to the conditional $P_{Y|\mathbf{X}=\mathbf{x}}$ instead of the marginal $P_Y$. This idea motivates a generalized version of the principle of maximum entropy, which we call the *principle of maximum conditional entropy*. In fact, this principle breaks Step 3 into two smaller steps:

3a. We search for $P^*$ the distribution maximizing the conditional entropy over $\Gamma(\hat{P})$,

3b. We find $\psi^*$ the optimal decision rule for $P^*$.

Although the principle of maximum conditional entropy characterizes the solution to (1), computing the maximizing distribution is hard in general. In [7], the authors propose a conditional version of the principle of maximum entropy, for the specific case of Shannon entropy, and draw the principle's connection to (1). They call it the principle of minimum mutual information, by which one should predict based on the distribution minimizing mutual information among $\mathbf{X}$ and $Y$. However, they develop their theory targeting a broad class of distribution sets, which results in a convex problem, yet the number of variables is exponential in the dimension of the problem.

To overcome this issue, we propose a specific structure for the distribution set by matching the marginal $P_{\mathbf{X}}$ of all the joint distributions $P_{\mathbf{X},Y}$ in $\Gamma(\hat{P})$ to the empirical marginal $\hat{P}_{\mathbf{X}}$ while matching only the cross-moments between $\mathbf{X}$ and $Y$ with those of the empirical distribution $\hat{P}_{\mathbf{X},\mathbf{Y}}$. We show that this choice of $\Gamma(\hat{P})$ has two key advantages: 1) the minimax decision rule $\psi^*$ can be computed efficiently; 2) the minimax generalization error can be controlled by allowing a level of uncertainty in the matching of the cross-moments, which can be viewed as regularization in the minimax framework. Our solution is achieved through convex duality. For some loss functions, the dual problem turns out to be equivalent to the maximum likelihood problem for generalized linear models. For example,

under quadratic and logarithmic loss functions this minimax approach revisits the linear and logistic regression models respectively.

On the other hand, for 0-1 loss, the minimax approach leads to a new randomized linear classifier which we call the *minimax SVM*. The minimax SVM minimizes the worst-case expected 0-1 loss over $\Gamma(\hat{P})$ by solving a tractable optimization problem. In contrast, the classic ERM formulation of minimizing the 0-1 loss over linear classifiers is well-known to be NP-hard [8]. Interestingly, the dual problem for the 0-1 loss minimax problem corresponds also to an ERM problem for linear classifiers, but with a loss function *different* from 0-1 loss. This loss function, which we call the *minimax-hinge loss*, is also different from the classic hinge loss (Figure 2). We emphasize that while the hinge loss is an *adhoc* surrogate loss function chosen to convexify the 0-1 loss ERM problem, the minimax-hinge loss *emerges* from the minimax formulation. We also perform several numerical experiments to demonstrate the power of the minimax SVM in outperforming the standard SVM which minimizes the surrogate hinge loss.

## 2 Principle of Maximum Conditional Entropy

In this section, we provide a conditional version of the key definitions and results developed in [5]. We propose the principle of maximum conditional entropy to break Step 3 into 3a and 3b in Figure 1. We also define and characterize Bayes decision rules for different loss functions to address Step 3b.

### 2.1 Decision Problems, Bayes Decision Rules, Conditional Entropy

Consider a decision problem. Here the decision maker observes $X \in \mathcal{X}$ from which she predicts a random target variable $Y \in \mathcal{Y}$ using an action $a \in \mathcal{A}$. Let $P_{X,Y} = (P_X, P_{Y|X})$ be the underlying distribution for the random pair $(X, Y)$. Given a loss function $L : \mathcal{Y} \times \mathcal{A} \to [0, \infty]$, $L(y, a)$ indicates the loss suffered by the decision maker by deciding action $a$ when $Y = y$. The decision maker uses a decision rule $\psi : \mathcal{X} \to \mathcal{A}$ to select an action $a = \psi(x)$ from $\mathcal{A}$ based on an observation $x \in \mathcal{X}$. We will in general allow the decision rules to be random, i.e. $\psi$ is random. The main purpose of extending to the space of randomized decision rules is to form a convex set of decision rules. Later in Theorem 1, this convexity is used to prove a saddle-point theorem.

We call a (randomized) decision rule $\psi_{\text{Bayes}}$ a Bayes decision rule if for all decision rules $\psi$ and for all $x \in \mathcal{X}$:
$$\mathbb{E}[L(Y, \psi_{\text{Bayes}}(X))|X = x] \leq \mathbb{E}[L(Y, \psi(X))|X = x].$$
It should be noted that $\psi_{\text{Bayes}}$ depends only on $P_{Y|X}$, i.e. it remains a Bayes decision rule under a different $P_X$. The (unconditional) entropy of $Y$ is defined as [5]
$$H(Y) := \inf_{a \in \mathcal{A}} \mathbb{E}[L(Y, a)]. \tag{2}$$

Similarly, we can define conditional entropy of $Y$ given $X = x$ as
$$H(Y|X = x) := \inf_{\psi} \mathbb{E}[L(Y, \psi(X))|X = x], \tag{3}$$

and the conditional entropy of $Y$ given $X$ as
$$H(Y|X) := \sum_x P_X(x) H(Y|X = x) = \inf_{\psi} \mathbb{E}[L(Y, \psi(X))]. \tag{4}$$

Note that $H(Y|X = x)$ and $H(Y|X)$ are both concave in $P_{Y|X}$. Applying Jensen's inequality, this concavity implies that
$$H(Y|X) \leq H(Y),$$
which motivates the following definition for the information that $X$ carries about $Y$,
$$I(X; Y) := H(Y) - H(Y|X), \tag{5}$$
i.e. the reduction of expected loss in predicting $Y$ by observing $X$. In [9], the author has defined the same concept to which he calls a coherent dependence measure. It can be seen that $I(X; Y) = \mathbb{E}_{P_X}[D(P_{Y|X}, P_Y)]$ where $D$ is the divergence measure corresponding to the loss $L$, defined for any two probability distributions $P_Y$, $Q_Y$ with Bayes actions $a_P$, $a_Q$ as [5]
$$D(P_Y, Q_Y) := E_P[L(Y, a_Q)] - E_P[L(Y, a_P)] = E_P[L(Y, a_Q)] - H_P(Y). \tag{6}$$

## 2.2 Examples

### 2.2.1 Logarithmic loss

For an outcome $y \in \mathcal{Y}$ and distribution $Q_Y$, define logarithmic loss as $L_{\log}(y, Q_Y) = -\log Q_Y(y)$. It can be seen $H_{\log}(Y)$, $H_{\log}(Y|X)$, $I_{\log}(X; Y)$ are the well-known unconditional, conditional Shannon entropy and mutual information [10]. Also, the Bayes decision rule for a distribution $P_{X,Y}$ is given by $\psi_{\text{Bayes}}(x) = P_{Y|X}(\cdot|x)$.

### 2.2.2 0-1 loss

The 0-1 loss function is defined for any $y, \hat{y} \in \mathcal{Y}$ as $L_{\text{0-1}}(y, \hat{y}) = \mathbb{I}(\hat{y} \neq y)$. Then, we can show

$$H_{\text{0-1}}(Y) = 1 - \max_{y \in \mathcal{Y}} P_Y(y), \quad H_{\text{0-1}}(Y|X) = 1 - \sum_{x \in \mathcal{X}} \max_{y \in \mathcal{Y}} P_{X,Y}(x, y).$$

The Bayes decision rule for a distribution $P_{X,Y}$ is the well-known maximum a posteriori (MAP) rule, i.e. $\psi_{\text{Bayes}}(x) = \text{argmax}_{y \in \mathcal{Y}} P_{Y|X}(y|x)$.

### 2.2.3 Quadratic loss

The quadratic loss function is defined as $L_2(y, \hat{y}) = (y - \hat{y})^2$. It can be seen

$$H_2(Y) = \text{Var}(Y), \quad H_2(Y|X) = \mathbb{E}\left[\text{Var}(Y|X)\right], \quad I_2(X; Y) = \text{Var}\left(\mathbb{E}[Y|X]\right).$$

The Bayes decision rule for any $P_{X,Y}$ is the well-known minimum mean-square error (MMSE) estimator that is $\psi_{\text{Bayes}}(x) = \mathbb{E}[Y|X = x]$.

## 2.3 Principle of Maximum Conditional Entropy & Robust Bayes decision rules

Given a distribution set $\Gamma$, consider the following minimax problem to find a decision rule minimizing the worst-case expected loss over $\Gamma$

$$\underset{\psi \in \mathbf{\Psi}}{\text{argmin}} \ \max_{P \in \Gamma} \mathbb{E}_P[L(Y, \psi(X))], \tag{7}$$

where $\mathbf{\Psi}$ is the space of all randomized mappings from $\mathcal{X}$ to $\mathcal{A}$ and $\mathbb{E}_P$ denotes the expected value over distribution $P$. We call any solution $\psi^*$ to the above problem a robust Bayes decision rule against $\Gamma$. The following results motivate a generalization of the maximum entropy principle to find a robust Bayes decision rule. Refer to the supplementary material for the proofs.

**Theorem 1.A.** *(Weak Version) Suppose $\Gamma$ is convex and closed, and let $L$ be a bounded loss function. Assume $\mathcal{X}, \mathcal{Y}$ are finite and that the risk set $S = \left\{ [L(y, a)]_{y \in \mathcal{Y}} : a \in \mathcal{A} \right\}$ is closed. Then there exists a robust Bayes decision rule $\psi^*$ against $\Gamma$, which is a Bayes decision rule for a distribution $P^*$ that maximizes the conditional entropy $H(Y|X)$ over $\Gamma$.*

**Theorem 1.B.** *(Strong Version) Suppose $\Gamma$ is convex and that under any $P \in \Gamma$ there exists a Bayes decision rule. We also assume the continuity in Bayes decision rules for distributions in $\Gamma$ (See the supplementary material for the exact condition). Then, if $P^*$ maximizes $H(Y|X)$ over $\Gamma$, any Bayes decision rule for $P^*$ is a robust Bayes decision rule against $\Gamma$.*

**Principle of Maximum Conditional Entropy**: Given a set of distributions $\Gamma$, predict $Y$ based on a distribution in $\Gamma$ that maximizes the conditional entropy of $Y$ given $X$, i.e.

$$\underset{P \in \Gamma}{\text{argmax}} \ H(Y|X) \tag{8}$$

Note that while the weak version of Theorem 1 guarantees **only the existence** of a saddle point for (7), the strong version further guarantees that **any** Bayes decision rule of the maximizing distribution results in a robust Bayes decision rule. However, the continuity in Bayes decision rules does not hold for the discontinuous 0-1 loss, which requires considering the weak version of Theorem 1 to address this issue.

# 3 Prediction via Maximum Conditional Entropy Principle

Consider a prediction task with target variable $Y$ and feature vector $\mathbf{X} = (X_1, \ldots, X_d)$. We do not require the variables to be discrete. As discussed earlier, the maximum conditional entropy principle reduces (7) to (8), which formulate steps 3 and 3a in Figure 1, respectively. However, a general formulation of (8) in terms of the joint distribution $P_{\mathbf{X},Y}$ leads to an exponential computational complexity in the feature dimension $d$.

The key question is therefore under what structures of $\Gamma(\hat{P})$ in Step 2 we can solve (8) efficiently. In this section, we propose a specific structure for $\Gamma(\hat{P})$, under which we provide an efficient solution to Steps 3a and 3b in Figure 1. In addition, we prove a bound on the excess worst-case risk for the proposed $\Gamma(\hat{P})$.

To describe this structure, consider a set of distributions $\Gamma(Q)$ centered around a given distribution $Q_{\mathbf{X},Y}$, where for a given norm $\|\cdot\|$, mapping vector $\boldsymbol{\theta}(Y)_{t \times 1}$,

$$\Gamma(Q) = \{ P_{\mathbf{X},Y} : P_{\mathbf{X}} = Q_{\mathbf{X}} , \tag{9}$$
$$\forall 1 \le i \le t : \ \| \mathbb{E}_P [\theta_i(Y)\mathbf{X}] - \mathbb{E}_Q [\theta_i(Y)\mathbf{X}] \| \le \epsilon_i \}.$$

Here $\boldsymbol{\theta}$ encodes $Y$ with $t$-dimensional $\boldsymbol{\theta}(Y)$, and $\theta_i(Y)$ denotes the $i$th entry of $\boldsymbol{\theta}(Y)$. The first constraint in the definition of $\Gamma(Q)$ requires all distributions in $\Gamma(Q)$ to share the same marginal on $\mathbf{X}$ as $Q$; the second imposes constraints on the cross-moments between $\mathbf{X}$ and $Y$, allowing for some uncertainty in estimation. When applied to the supervised learning problem, we will choose $Q$ to be the empirical distribution $\hat{P}$ and select $\boldsymbol{\theta}$ appropriately based on the loss function $L$. However, for now we will consider the problem of solving (8) over $\Gamma(Q)$ for general $Q$ and $\boldsymbol{\theta}$.

To that end, we use a similar technique as in the Fenchel's duality theorem, also used at [11, 12, 13] to address divergence minimization problems. However, we consider a different version of convex conjugate for $-H$, which is defined with respect to $\boldsymbol{\theta}$. Considering $\mathcal{P}_\mathcal{Y}$ as the set of all probability distributions for the variable $Y$, we define $F_{\boldsymbol{\theta}} : \mathbb{R}^t \to \mathbb{R}$ as the convex conjugate of $-H(Y)$ with respect to the mapping $\boldsymbol{\theta}$,

$$F_{\boldsymbol{\theta}}(\mathbf{z}) := \max_{P \in \mathcal{P}_\mathcal{Y}} H(Y) + \mathbb{E}[\boldsymbol{\theta}(Y)]^T \mathbf{z}. \tag{10}$$

**Theorem 2.** *Define $\Gamma(Q)$, $F_{\boldsymbol{\theta}}$ as given by (9), (10). Then the following duality holds*

$$\max_{P \in \Gamma(Q)} H(Y|\mathbf{X}) = \min_{\mathbf{A} \in \mathbb{R}^{t \times d}} \mathbb{E}_Q \left[ F_{\boldsymbol{\theta}}(\mathbf{A}\mathbf{X}) - \boldsymbol{\theta}(Y)^T \mathbf{A}\mathbf{X} \right] + \sum_{i=1}^{t} \epsilon_i \|\mathbf{A}_i\|_*, \tag{11}$$

*where $\|\mathbf{A}_i\|_*$ denotes $\|\cdot\|$'s dual norm of the $\mathbf{A}$'s $i$th row. Furthermore, for the optimal $P^*$ and $\mathbf{A}^*$*

$$\mathbb{E}_{P^*}[\boldsymbol{\theta}(Y) \,|\, \mathbf{X} = \mathbf{x}] = \nabla F_{\boldsymbol{\theta}}(\mathbf{A}^*\mathbf{x}). \tag{12}$$

*Proof.* Refer to the the supplementary material for the proof. $\qquad\square$

When applying Theorem 2 on a supervised learning problem with a specific loss function, $\theta$ will be chosen such that $\mathbb{E}_{P^*}[\boldsymbol{\theta}(Y) \,|\, \mathbf{X} = \mathbf{x}]$ provides sufficient information to compute the Bayes decision rule $\Psi^*$ for $P^*$. This enables the direct computation of $\psi^*$, i.e. step 3 of Figure 1, without the need to explicitly compute $P^*$ itself. For the loss functions discussed at Subsection 2.2, we choose the identity $\boldsymbol{\theta}(Y) = Y$ for the quadratic loss and the one-hot encoding $\boldsymbol{\theta}(Y) = [\mathbb{I}(Y = i)]_{i=1}^{t}$ for the logarithmic and 0-1 loss functions. Later in this section, we will discuss how this theorem applies to these loss functions.

## 3.1 Generalization Bounds for the Worst-case Risk

By establishing the objective's Lipschitzness and boundedness through appropriate assumptions, we can bound the rate of uniform convergence for the problem in the RHS of (11) [14]. Here we consider the uniform convergence of the empirical averages, when $Q = \hat{P}_n$ is the empirical distribution of $n$ samples drawn i.i.d. from the underlying distribution $\tilde{P}$, to their expectations when $Q = \tilde{P}$.

In the supplementary material, we prove the following theorem which bounds the excess worst-case risk. Here $\hat{\psi}_n$ and $\tilde{\psi}$ denote the robust Bayes decision rules against $\Gamma(\hat{P}_n)$ and $\Gamma(\tilde{P})$, respectively.

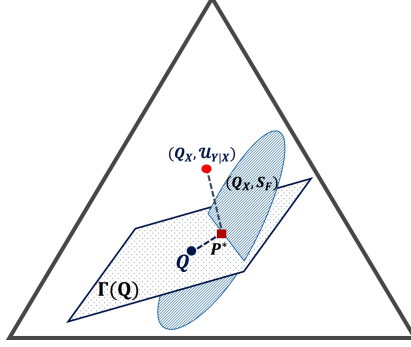

Figure 3: Duality of Maximum Conditional Entropy/Maximum Likelihood in GLMs

As explained earlier, by the maximum conditional entropy principle we can learn $\hat{\psi}_n$ by solving the RHS of (11) for the empirical distribution of samples and then applying (12).

**Theorem 3.** *Consider a loss function $L$ with the entropy function $H$ and suppose $\boldsymbol{\theta}(Y)$ includes only one element, i.e. $t = 1$. Let $M = \max_{P \in \mathcal{P}_\mathcal{Y}} H(Y)$ be the maximum entropy value over $\mathcal{P}_\mathcal{Y}$. Also, take $\| \cdot \| / \| \cdot \|_*$ to be the $\ell_p / \ell_q$ pair where $\frac{1}{p} + \frac{1}{q} = 1$, $1 \le q \le 2$. Given that $\|\mathbf{X}\|_2 \le B$ and $|\theta(Y)| \le L$, for any $\delta > 0$ with probability at least $1 - \delta$*

$$\max_{P \in \Gamma(\tilde{P})} \mathbb{E}[L(Y, \hat{\psi}_n(\mathbf{X}))] - \max_{P \in \Gamma(\tilde{P})} \mathbb{E}[L(Y, \tilde{\psi}(\mathbf{X}))] \le \frac{4BLM}{\epsilon \sqrt{n}} \left( 1 + \sqrt{\frac{9 \log(4/\delta)}{8}} \right). \quad (13)$$

Theorem 3 states that though we learn the prediction rule $\hat{\psi}_n$ by solving the maximum conditional problem for the empirical case, we can bound the excess $\Gamma$-based worst-case risk. This result justifies the specific constraint of fixing the marginal $P_\mathbf{X}$ across the proposed $\Gamma(Q)$ and explains the role of the uncertainty parameter $\epsilon$ in bounding the excess worst-case risk.

### 3.2 A Minimax Interpretation of Generalized Linear Models

We make the key observation that if $F_{\boldsymbol{\theta}}$ is the log-partition function of an expoenetial-family distribution, the problem in the RHS of (11), when $\epsilon_i = 0$ for all $i$'s, is equivalent to minimizing the negative log-likelihood for fitting a generalized linear model [15] given by

- An exponential-family distribution $p(y|\boldsymbol{\eta}) = h(y) \exp \left( \boldsymbol{\eta}^T \boldsymbol{\theta}(y) - F_{\boldsymbol{\theta}}(\boldsymbol{\eta}) \right)$ with the log-partition function $F_{\boldsymbol{\theta}}$ and the sufficient statistic $\boldsymbol{\theta}(Y)$,
- A linear predictor, $\boldsymbol{\eta}(\mathbf{X}) = \mathbf{A}\mathbf{X}$,
- A mean function, $\mathbb{E}[\boldsymbol{\theta}(Y)|\mathbf{X} = \mathbf{x}] = \nabla F_{\boldsymbol{\theta}}(\boldsymbol{\eta}(\mathbf{x}))$.

Therefore, Theorem 2 reveals a duality between the maximum conditional entropy problem over $\Gamma(Q)$ and the regularized maximum likelihood problem for the specified generalized linear model.

As a geometric interpretation of this duality, by solving the regularized maximum likelihood problem in the RHS of (11), we in fact minimize a regularized KL-divergence

$$\operatorname*{argmin}_{P_{Y|\mathbf{X}} \in S_F} \mathbb{E}_{Q_\mathbf{X}}[ D_{\mathrm{KL}}( Q_{Y|\mathbf{X}} \| P_{Y|\mathbf{X}} )] + \sum_{i=1}^{t} \epsilon_i \|\mathbf{A}_i(P_{Y|\mathbf{X}})\|_*, \quad (14)$$

where $S_F = \{P_{Y|\mathbf{X}}(y|\mathbf{x}) = h(y) \exp( \boldsymbol{\theta}(y)^T \mathbf{A}\mathbf{x} - F_{\boldsymbol{\theta}}(\mathbf{A}\mathbf{x}) \mid \mathbf{A} \in \mathbb{R}^{t \times s}\}$ is the set of all exponential-family conditional distributions for the specified generalized linear model. This can be viewed as projecting $Q$ onto $(Q_X, S_F)$ (See Figure 3).

Furthermore, for a label-invariant entropy $H(Y)$ the Bayes act for the uniform distribution $\mathcal{U}_Y$ leads to the same expected loss under any distribution on $Y$. Based on the divergence $D$'s definition in (6), maximizing $H(Y|\mathbf{X})$ over $\Gamma(Q)$ in the LHS of (11) is therefore equivalent to the following divergence minimization problem

$$\operatorname*{argmin}_{P_{Y|\mathbf{X}}: (Q_\mathbf{X}, P_{Y|\mathbf{x}}) \in \Gamma(Q)} \mathbb{E}_{Q_\mathbf{X}}[ D(P_{Y|\mathbf{X}}, \mathcal{U}_{Y|\mathbf{X}})]. \quad (15)$$

Here $\mathcal{U}_{Y|\mathbf{X}}$ denotes the uniform conditional distribution over $Y$ given any $x \in \mathcal{X}$. This can be interpreted as projecting the joint distribution $(Q_{\mathbf{X}}, \mathcal{U}_{Y|\mathbf{X}})$ onto $\Gamma(Q)$ (See Figure 3). Then, the duality shown in Theorem 2 implies the following corollary.

**Corollary 1.** *Given a label-invariant H, the solution to (14) also minimizes (15), i.e. (14) $\subseteq$ (15).*

### 3.3 Examples

#### 3.3.1 Logarithmic Loss: Logistic Regression

To gain sufficient information for the Bayes decision rule under the logarithmic loss, for $Y \in \mathcal{Y} = \{1, \ldots, t+1\}$, let $\boldsymbol{\theta}(Y)$ be the one-hot encoding of $Y$, i.e. $\boldsymbol{\theta}_i(Y) = \mathbb{I}(Y = i)$ for $1 \leq i \leq t$. Here, we exclude $i = t+1$ as $\mathbb{I}(Y = t+1) = 1 - \sum_{i=1}^{t} \mathbb{I}(Y = i)$. Then

$$F_{\boldsymbol{\theta}}(\mathbf{z}) = \log\big(1 + \sum_{j=1}^{t} \exp(\mathbf{z}_j)\big), \quad \forall 1 \leq i \leq t: \ \big(\nabla F_{\boldsymbol{\theta}}(\mathbf{z})\big)_i = \exp(\mathbf{z}_i)/\big(1 + \sum_{j=1}^{t} \exp(\mathbf{z}_j)\big), \quad (16)$$

which is the logistic regression model [16]. Also, the RHS of (11) will be the regularized maximum likelihood problem for logistic regression. This particular result is well-studied in the literature and straightforward using the duality shown in [17].

#### 3.3.2 0-1 Loss: Minimax SVM

To get sufficient information for the Bayes decision rule under the 0-1 loss, we again consider the one-hot encoding $\boldsymbol{\theta}$ described for the logarithmic loss. We show in the supplementary material that if $\tilde{\mathbf{z}} = (\mathbf{z}, 0)$ and $\tilde{z}_{(i)}$ denotes the $i$th largest element of $\tilde{\mathbf{z}}$,

$$F_{\boldsymbol{\theta}}(\mathbf{z}) = \max_{1 \leq k \leq t+1} \ \frac{k - 1 + \sum_{j=1}^{k} \tilde{z}_{(j)}}{k}. \quad (17)$$

In particular, if $Y \in \mathcal{Y} = \{-1, 1\}$ is binary the dual problem (11) for learning the optimal linear predictor $\boldsymbol{\alpha}^*$ given $n$ samples $(\mathbf{x}_i, y_i)_{i=1}^{n}$ will be

$$\min_{\boldsymbol{\alpha}} \ \frac{1}{n} \sum_{i=1}^{n} \max\left\{0, \frac{1 - y_i \boldsymbol{\alpha}^T \mathbf{x}_i}{2}, -y_i \boldsymbol{\alpha}^T \mathbf{x}_i\right\} + \epsilon \|\boldsymbol{\alpha}\|_*. \quad (18)$$

The first term is the empirical risk of a linear classifier over the minimax-hinge loss $\max\{0, \frac{1-z}{2}, -z\}$ as shown in Figure 2. In contrast, the standard SVM is formulated using the hinge loss $\max\{0, 1-z\}$:

$$\min_{\boldsymbol{\alpha}} \ \frac{1}{n} \sum_{i=1}^{n} \max\{0, 1 - y_i \boldsymbol{\alpha}^T \mathbf{x}_i\} + \epsilon \|\boldsymbol{\alpha}\|_*, \quad (19)$$

We therefore call this classification approach the minimax SVM. However, unlike the standard SVM, the minimax SVM is naturally extended to multi-class classification.

Using Theorem 1.A[2], we prove that for 0-1 loss the robust Bayes decision rule exists and is randomized in general, where given the optimal linear predictor $\tilde{\mathbf{z}} = (\mathbf{A}^* \mathbf{x}, 0)$ randomly predicts a label according to the following $\tilde{\mathbf{z}}$-based distribution on labels

$$\forall 1 \leq i \leq t+1: \quad p_{\sigma(i)} = \begin{cases} \tilde{z}_{(i)} + \dfrac{1 - \sum_{j=1}^{k_{\max}} \tilde{z}_{(j)}}{k_{\max}} & \text{if } \sigma(i) \leq k_{\max}, \\ 0 & \text{Otherwise.} \end{cases} \quad (20)$$

Here $\sigma$ is the permutation sorting $\tilde{\mathbf{z}}$ in the ascending order, i.e. $\tilde{z}_{\sigma(i)} = \tilde{z}_{(i)}$, and $k_{\max}$ is the largest index $k$ satisfying $\sum_{i=1}^{k} [\tilde{\mathbf{z}}_{(i)} - \tilde{\mathbf{z}}_{(k)}] < 1$. For example, in the binary case discussed, the minimax SVM first solves (18) to find the optimal $\boldsymbol{\alpha}^*$ and then predicts label $y = 1$ vs. label $y = -1$ with probability $\min\{1, \max\{0, (1 + \mathbf{x}^T \boldsymbol{\alpha}^*)/2\}\}$.

| Dataset | mmSVM | SVM | DCC | MPM | TAN | DRC |
|---------|-------|-----|-----|-----|-----|-----|
| adult | **17** | 22 | 18 | 22 | **17** | **17** |
| credit | **12** | 16 | 14 | 13 | 17 | 13 |
| kr-vs-kp | 4 | **3** | 10 | 5 | 7 | 5 |
| promoters | **5** | 9 | **5** | 6 | 44 | 6 |
| votes | **3** | 5 | **3** | 4 | 8 | **3** |
| hepatitis | **17** | 20 | 19 | 18 | **17** | **17** |

Table 1: Methods Performance (error in %)

### 3.3.3 Quadratic Loss: Linear Regression

Based on the Bayes decision rule for the quadratic loss, we choose $\boldsymbol{\theta}(Y) = Y$. To derive $F_{\boldsymbol{\theta}}$, note that if we let $\mathcal{P}_{\mathcal{Y}}$ in (10) include all possible distributions, the maximized entropy (variance for quadratic loss) and thus the value of $F_{\boldsymbol{\theta}}$ would be infinity. Therefore, given a parameter $\rho$, we restrict the second moment of distributions in $\mathcal{P}_{\mathcal{Y}} = \{P_Y : \mathbb{E}[Y^2] \leq \rho^2\}$ and then apply (10). We show in the supplementary material that an adjusted version of Theorem 2 holds after this change, and

$$F_{\boldsymbol{\theta}}(z) - \rho^2 = \begin{cases} z^2/4 & \text{if } |z/2| \leq \rho \\ \rho(|z| - \rho) & \text{if } |z/2| > \rho, \end{cases} \tag{21}$$

which is the Huber function [18]. Given the samples of a supervised learning task if we choose the parameter $\rho$ large enough, by solving the RHS of (11) when $F_{\boldsymbol{\theta}}(z)$ is replaced with $z^2/4$ and set $\rho$ greater than $\max_i |\mathbf{A}^* \mathbf{x}_i|$, we can equivalently take $F_{\boldsymbol{\theta}}(z) = z^2/4 + \rho^2$. Then, by (12) we derive the linear regression model and the RHS of (11) is equivalent to

– Least squares when $\epsilon = 0$.
– Lasso [19, 20] when $\|\cdot\|/\|\cdot\|_*$ is the $\ell_\infty/\ell_1$ pair.
– Ridge regression [21] when $\|\cdot\|$ is the $\ell_2$-norm.
– (overlapping) Group lasso [22, 23] with the $\ell_{1,p}$ penalty when $\Gamma_{\mathrm{GL}}(Q)$ is defined, given subsets $I_1, \ldots I_k$ of $\{1, \ldots, d\}$ and $1/p + 1/q = 1$, as

$$\Gamma_{\mathrm{GL}}(Q) = \{ P_{\mathbf{X},Y} : P_{\mathbf{X}} = Q_{\mathbf{X}}, \tag{22}$$
$$\forall 1 \leq j \leq k : \| \mathbb{E}_P [Y\mathbf{X}_{I_j}] - \mathbb{E}_Q [Y\mathbf{X}_{I_j}] \|_q \leq \epsilon_j \}.$$

## 4 Numerical Experiments

We evaluated the performance of the minimax SVM on six binary classification datasets from the UCI repository, compared to these five benchmarks: Support Vector Machines (SVM) [24], Discrete Chebyshev Classifiers (DCC) [3], Minimax Probabilistic Machine (MPM) [2], Tree Augmented Naive Bayes (TAN) [25], and Discrete Rényi Classifiers (DRC) [4]. The results are summarized in Table 1 where the numbers indicate the percentage of error in the classification task.

We implemented the minimax SVM by applying the subgradient descent to (18) with the regularizer $\lambda \|\boldsymbol{\alpha}\|_2^2$. We determined the parameters by cross validation, where we used a randomly-selected 70% of the training set for training and the rest 30% for testing. We tested the values in $\{2^{-10}, \ldots, 2^{10}\}$. Using the tuned parameters, we trained the algorithm over all the training set and then evaluated the error rate over the test set. We performed this procedure in 1000 Monte Carlo runs each training on 70% of the data points and testing on the rest 30% and averaged the results.

As seen in the table, the minimax SVM results in the best performance for five of the six datasets. To compare these methods in high-dimensional problems, we ran an experiment over synthetic data with $n = 200$ samples and $d = 10000$ features. We generated features by i.i.d. Bernoulli with $P(X_i = 1) = 0.7$, and considered $y = \text{sign}(\gamma^T \mathbf{x} + z)$ where $z \sim N(0, 1)$. Using the above procedure, we evaluated 19.3% for the mmSVM, 19.5% error rate for SVM, 19.6% error rate for DRC, which indicates the mmSVM can outperform SVM and DRC in high-dimensional settings as well. Also, the average training time for training mmSVM was 0.085 seconds, faster than the training time for the SVM (using Matlab's SVM command) with the average 0.105 seconds.

**Acknowledgments:** We are grateful to Stanford University providing a Stanford Graduate Fellowship, and the Center for Science of Information (CSoI), an NSF Science and Technology Center under grant agreement CCF-0939370 , for the support during this research.

## Footnotes

[2] We show that given the specific structure of $\Gamma(Q)$ Theorem 1.A holds whether $\mathcal{X}$ is finite or infinite.

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
