[Supplementary Material]

# Supplementary Material:
# A Minimax Approach to Supervised Learning

**Farzan Farnia, David Tse**
Department of Electrical Engineering
Stanford University
{farnia,dntse}@stanford.edu

## 1   Proof of Theorem 1

### 1.a   Weak Version

First, we list the assumptions of the weak version of Theorem 1:

- $\Gamma$ is convex and closed,
- Loss function $L$ is bounded by a constant $C$,
- $\mathcal{X}, \mathcal{Y}$ are finite,
- Risk set $S = \left\{ [L(y,a)]_{y \in \mathcal{Y}} : a \in \mathcal{A} \right\}$ is closed.

Given these assumptions, Sion's minimax theorem [1] implies that the minimax problem has a finite answer $H^*$,

$$H^* := \sup_{P \in \Gamma} \inf_{\psi \in \Psi} \mathbb{E}[L(Y, \psi(X))] = \inf_{\psi \in \Psi} \sup_{P \in \Gamma} \mathbb{E}[L(Y, \psi(X))]. \tag{1}$$

Thus, there exists a sequence of decision rules $(\psi_n)_{n=1}^{\infty}$ for which

$$\lim_{n \to \infty} \sup_{P \in \Gamma} \mathbb{E}[L(Y, \psi_n(X))] = H^*. \tag{2}$$

As we supposed, the risk set $S$ is closed. Therefore, the randomized risk set[1] $S_r = \left\{ [L(y,\zeta)]_{y \in \mathcal{Y}} : \zeta \in \mathcal{Z} \right\}$ defined over the space of randomized acts $\mathcal{Z}$ is also closed and, since $L$ is bounded, is a compact subset of $\mathbb{R}^{|\mathcal{Y}|}$. Therefore, since $\mathcal{X}$ and $\mathcal{Y}$ are both finite, we can find a randomized decision rule $\psi^*$ which on taking a subsequence $(n_k)_{k=1}^{\infty}$ satisfies

$$\forall x \in \mathcal{X}, y \in \mathcal{Y}: \quad L(y, \psi^*(x)) = \lim_{k \to \infty} L(y, \psi_{n_k}(x)). \tag{3}$$

Then $\psi^*$ is a robust Bayes decision rule against $\Gamma$, because

$$\sup_{P \in \Gamma} \mathbb{E}\left[L(Y, \psi^*(X))\right] = \sup_{P \in \Gamma} \lim_{k \to \infty} \mathbb{E}\left[L(Y, \psi_{n_k}(X))\right] \leq \lim_{k \to \infty} \sup_{P \in \Gamma} \mathbb{E}[L(Y, \psi_{n_k}(X))] = H^*. \tag{4}$$

Moreover, since $\Gamma$ is assumed to be convex and closed (hence compact), $H(Y|X)$ achieves its supremum over $\Gamma$ at some distribution $P^*$. By the definition of conditional entropy, (4) implies that

$$E_{P^*}[L(Y, \psi^*(X))] \leq \sup_{P \in \Gamma} \mathbb{E}\left[L(Y, \psi^*(X))\right] \leq H^* = H_{P^*}(Y|X), \tag{5}$$

which shows that $\psi^*$ is a Bayes decision rule for $P^*$ as well. This completes the proof.

## 1.b Strong Version

Let's recall the assumptions of the strong version of Theorem 1:

- $\Gamma$ is convex.
- For any distribution $P \in \Gamma$, there exists a Bayes decision rule.
- We assume continuity in Bayes decision rules over $\Gamma$, i.e., if a sequence of distributions $(Q_n)_{n=1}^{\infty} \in \Gamma$ with the corresponding Bayes decision rules $(\psi_n)_{n=1}^{\infty}$ converges to $Q$ with a Bayes decision rule $\psi$, then under any $P \in \Gamma$, the expected loss of $\psi_n$ converges to the expected loss of $\psi$.
- $P^*$ maximizes the conditional entropy $H(Y|X)$.

**Note:** A particular structure used in our paper is given by fixing the marginal $P_X$ across $\Gamma$. Under this structure, the condition of the continuity in Bayes decision rules reduces to the continuity in Bayes acts over $P_Y$'s in $\Gamma_{Y|X}$. It can be seen that while this condition holds for the logarithmic and quadratic loss functions, it does not hold for the 0-1 loss.

Let $\psi^*$ be a Bayes decision rule for $P^*$. We need to show that $\psi^*$ is a robust Bayes decision rule against $\Gamma$. To show this, it suffices to show that $(P^*, \psi^*)$ is a saddle point of the mentioned minimax problem, i.e.,

$$\mathbb{E}_{P^*}[L(Y, \psi^*(X))] \leq \mathbb{E}_{P^*}[L(Y, \psi(X))], \tag{6}$$

and

$$\mathbb{E}_{P^*}[L(Y, \psi^*(X))] \geq \mathbb{E}_{P}[L(Y, \psi^*(X))]. \tag{7}$$

Clearly, inequality (6) holds due to the definition of the Bayes decision rule. To show (7), let us fix an arbitrary distribution $P \in \Gamma$. For any $\lambda \in (0, 1]$, define $P_\lambda = \lambda P + (1 - \lambda)P^*$. Notice that $P_\lambda \in \Gamma$ since $\Gamma$ is convex. Let $\psi_\lambda$ be a Bayes decision rule for $P_\lambda$. Due to the linearity of the expected loss in the probability distribution, we have

$$\begin{aligned}
\mathbb{E}_{P}[L(Y, \psi_\lambda(X))] - \mathbb{E}_{P^*}[L(Y, \psi_\lambda(X))] &= \frac{\mathbb{E}_{P_\lambda}[L(Y, \psi_\lambda(X))] - \mathbb{E}_{P^*}[L(Y, \psi_\lambda(X))]}{\lambda} \\
&\leq \frac{H_{P_\lambda}(Y|X) - H_{P^*}(Y|X)}{\lambda} \\
&\leq 0,
\end{aligned}$$

for any $0 < \lambda \leq 1$. Here the first inequality is due to the definition of the conditional entropy and the last inequality holds since $P^*$ maximizes the conditional entropy over $\Gamma$. Applying the assumption of the continuity in Bayes decision rules, we have

$$\mathbb{E}_{P}[L(Y, \psi^*(X))] - \mathbb{E}_{P^*}[L(Y, \psi^*(X))] = \lim_{\lambda \to 0} \mathbb{E}_{P}[L(Y, \psi_\lambda(X))] - \mathbb{E}_{P^*}[L(Y, \psi_\lambda(X))] \leq 0, \tag{8}$$

which makes the proof complete.

## 2 Proof of Theorem 2

Let us recall the definition of the set $\Gamma(Q)$:

$$\begin{aligned}
\Gamma(Q) = \{ P_{\mathbf{X},Y} : P_{\mathbf{X}} = Q_{\mathbf{X}} , \qquad\qquad\qquad\qquad\qquad\qquad\qquad\qquad &\\
\forall 1 \leq i \leq t : \ \| \mathbb{E}_P[\theta_i(Y)\mathbf{X}] - \mathbb{E}_Q[\theta_i(Y)\mathbf{X}] \| \leq \epsilon_i \}. &
\end{aligned} \tag{9}$$

Defining $\tilde{\mathbf{E}}_i \triangleq \mathbb{E}_Q[\theta_i(Y)\mathbf{X}]$ and $C_i \triangleq \{\mathbf{u} : \|\mathbf{u} - \tilde{\mathbf{E}}_i\| \leq \epsilon_i\}$, we have

$$\max_{P \in \Gamma(Q)} H(Y|\mathbf{X}) = \max_{P, \mathbf{w}: \ \forall i: \ \mathbf{w}_i = \mathbb{E}_P[\theta_i(Y)\mathbf{X}]} \mathbb{E}_{Q_{\mathbf{X}}}[H_P(Y|\mathbf{X} = \mathbf{x})] + \sum_{i=1}^{t} I_{C_i}(\mathbf{w}_i) \tag{10}$$

where $I_C$ is the indicator function for the set $C$ defined as

$$I_C(x) = \begin{cases} 0 & \text{if } x \in C, \\ -\infty & \text{Otherwise.} \end{cases} \tag{11}$$

First of all, the law of iterated expectations implies that $\mathbb{E}_P\left[\theta_i(Y)\mathbf{X}\right] = \mathbb{E}_{Q_{\mathbf{X}}}\left[\mathbf{X}\,\mathbb{E}[\theta_i(Y)|\mathbf{X} = \mathbf{x}]\right]$.

Furthermore, (10) is equivalent to a convex optimization problem where it is not hard to check that the Slater condition is satisfied. Hence strong duality holds and we can write the dual problem as

$$\min_{\mathbf{A}} \ \sup_{P_{Y|\mathbf{X}},\mathbf{w}} \ \mathbb{E}_{Q_{\mathbf{X}}}\left[H_P(Y|\mathbf{X} = \mathbf{x}) + \sum_{i=1}^{t}\mathbb{E}[\theta_i(Y)|\mathbf{X} = \mathbf{x}]\mathbf{A}_i\mathbf{X}\right] + \sum_{i=1}^{t}\left[I_{C_i}(\mathbf{w}_i) - \mathbf{A}_i\mathbf{w}_i\right], \qquad (12)$$

where the rows of matrix $\mathbf{A}$, denoted by $\mathbf{A}_i$, are the Lagrange multipliers for the constraints of $\mathbf{w}_i = \mathbb{E}_P\left[\theta_i(Y)\mathbf{X}\right]$. Notice that the above problem decomposes across $P_{Y|\mathbf{X}=\mathbf{x}}$'s and $\mathbf{w}_i$'s. Hence, the dual problem can be rewritten as

$$\min_{\mathbf{A}} \left[\mathbb{E}_{Q_{\mathbf{X}}}\left[\sup_{P_{Y|\mathbf{X}=\mathbf{x}}} H_P(Y|\mathbf{X} = \mathbf{x}) + \sum_{i=1}^{t}\mathbb{E}[\theta_i(Y)|\mathbf{X} = \mathbf{x}]\mathbf{A}_i\mathbf{X}\right] + \sum_{i=1}^{t}\sup_{\mathbf{w}_i}\left[I_{C_i}(\mathbf{w}_i) - \mathbf{A}_i\mathbf{w}_i\right]\right] \qquad (13)$$

Furthermore, according to the definition of $F_{\boldsymbol{\theta}}$, we have

$$F_{\boldsymbol{\theta}}(\mathbf{A}\mathbf{x}) = \sup_{P_{Y|\mathbf{X}=\mathbf{x}}} H(Y|\mathbf{X} = \mathbf{x}) + \mathbb{E}[\boldsymbol{\theta}(Y)|\mathbf{X} = \mathbf{x}]^T\mathbf{A}\mathbf{x}. \qquad (14)$$

Moreover, the definition of the dual norm $\|\cdot\|_*$ implies

$$\sup_{\mathbf{w}_i} I_{C_i}(\mathbf{w}_i) - \mathbf{A}_i\mathbf{w}_i = \max_{\mathbf{u}\in C_i} -\mathbf{A}_i\mathbf{u} = -\mathbf{A}_i\tilde{\mathbf{E}}_i + \epsilon_i\|\mathbf{A}_i\|_*. \qquad (15)$$

Plugging (14) and (15) in (13), the dual problem can be simplified to

$$\min_{\mathbf{A}} \ \mathbb{E}_{Q_{\mathbf{X}}}\left[F_{\boldsymbol{\theta}}(\mathbf{A}\mathbf{X}) - \sum_{i=1}^{t}\mathbf{A}_i\tilde{\mathbf{E}}_i\right] + \sum_{i=1}^{t}\epsilon_i\|\mathbf{A}_i\|_*$$

$$= \min_{\mathbf{A}} \ \mathbb{E}_Q\left[F_{\boldsymbol{\theta}}(\mathbf{A}\mathbf{X}) - \boldsymbol{\theta}(Y)^T\mathbf{A}\mathbf{X}\right] + \sum_{i=1}^{t}\epsilon_i\|\mathbf{A}_i\|_*, \qquad (16)$$

which is equal to the primal problem (10) since the strong duality holds. Furthermore, note that we can rewrite the definition given for $F_{\boldsymbol{\theta}}$ as

$$F_{\boldsymbol{\theta}}(\mathbf{z}) = \max_{\mathbf{E}\in\mathbb{R}^t} G(\mathbf{E}) + \mathbf{E}^T\mathbf{z}, \qquad (17)$$

where we define

$$G(\mathbf{E}) = \begin{cases} \max\limits_{P\in\mathcal{P}_{\mathcal{Y}}:\,\mathbb{E}[\boldsymbol{\theta}(Y)]=\mathbf{E}} H(Y) & \text{if } \{P\in\mathcal{P}_{\mathcal{Y}}:\mathbb{E}[\boldsymbol{\theta}(Y)]=\mathbf{E}\}\neq\emptyset \\ -\infty & \text{Otherwise.} \end{cases} \qquad (18)$$

Observe that $F_{\boldsymbol{\theta}}$ is the convex conjugate of the convex $-G$. Therefore, applying the derivative property of convex conjugates [2] to (14),

$$\mathbb{E}_{P^*}[\boldsymbol{\theta}(Y)\,|\,\mathbf{X} = \mathbf{x}] \in \partial F_{\boldsymbol{\theta}}(\mathbf{A}^*\mathbf{x}). \qquad (19)$$

Here, $\partial F_{\boldsymbol{\theta}}$ denotes the subgradient of $F_{\boldsymbol{\theta}}$. Assuming $F_{\boldsymbol{\theta}}$ is differentiable at $\mathbf{A}^*\mathbf{x}$, (19) implies that

$$\mathbb{E}_{P^*}[\boldsymbol{\theta}(Y)\,|\,\mathbf{X} = \mathbf{x}] = \nabla F_{\boldsymbol{\theta}}(\mathbf{A}^*\mathbf{x}). \qquad (20)$$

## 2.a A generalization of Theorem 2

It can be seen that the above proof can be slightly generalized to prove the following generalization of Theorem 2.

**Theorem.** *Given a conjugate pair of convex functions $g$, $g^*$, the following duality holds*

$$\max_{P:\,P_X=Q_X} H(Y|\mathbf{X}) - \sum_{i=1}^{t} g\left(\mathbb{E}_P[\theta_i(Y)\mathbf{X}] - \mathbb{E}_Q[\theta_i(Y)\mathbf{X}]\right) = \qquad (21)$$

$$\min_{\mathbf{A}\in\mathbb{R}^{\mathbf{t}\times\mathbf{d}}} \ \mathbb{E}_Q\left[F_{\boldsymbol{\theta}}(\mathbf{A}\mathbf{X}) - \boldsymbol{\theta}(Y)^T\mathbf{A}\mathbf{X}\right] + \sum_{i=1}^{t} g^*(\mathbf{A}_i), \qquad (22)$$

*where $\mathbf{A}_i$ denotes the $i$th row of $\mathbf{A}$. In addition, for the optimal $P^*$ and $\mathbf{A}^*$*

$$\mathbb{E}_{P^*}[\boldsymbol{\theta}(Y)\,|\,\mathbf{X} = \mathbf{x}] = \nabla F_{\boldsymbol{\theta}}(\mathbf{A}^*\mathbf{x}). \qquad (23)$$

**Corollary.** *Consider a pair of dual norms $\|\cdot\|$, $\|\cdot\|_*$. Then, the following duality holds*

$$\max_{P:\,P_X=Q_X} H(Y|\mathbf{X}) - \sum_{i=1}^{t} \frac{1}{2\lambda_i} \left\| \mathbb{E}_P[\theta_i(Y)\mathbf{X}] - \mathbb{E}_Q[\theta_i(Y)\mathbf{X}] \right\|^2 = \tag{24}$$

$$\min_{\mathbf{A}\in\mathbb{R}^{\mathbf{t}\times\mathbf{d}}} \mathbb{E}_Q\left[ F_{\boldsymbol{\theta}}(\mathbf{A}\mathbf{X}) - \boldsymbol{\theta}(Y)^T\mathbf{A}\mathbf{X} \right] + \sum_{i=1}^{t} \frac{\lambda_i}{2} \|\mathbf{A}_i\|_*^2, \tag{25}$$

*where $\lambda_i$'s are positive real numbers and $\mathbf{A}_i$ denotes the $i$th row of $\mathbf{A}$. Moreover, for the optimal $P^*$ and $\mathbf{A}^*$*

$$\mathbb{E}_{P^*}[\boldsymbol{\theta}(Y)\,|\,\mathbf{X}=\mathbf{x}] = \nabla F_{\boldsymbol{\theta}}(\mathbf{A}^*\mathbf{x}). \tag{26}$$

## 3 Proof of Theorem 3

First, we aim to show that

$$\max_{P\in\Gamma(\tilde{P})} \mathbb{E}[L(Y,\hat{\psi}_n(\mathbf{X}))] \le \mathbb{E}_{\tilde{P}}\left[ F_{\boldsymbol{\theta}}(\hat{\mathbf{A}}_n\mathbf{X}) - \boldsymbol{\theta}(Y)^T\hat{\mathbf{A}}_n\mathbf{X} \right] + \sum_{i=1}^{t} \epsilon_i \|\hat{\mathbf{A}}_{n_i}\|_* \tag{27}$$

where $\hat{\mathbf{A}}_n$ denotes the solution to the RHS of the duality equation in Theorem 2 for the empirical distribution $\hat{P}_n$. Similar to the duality proven in Theorem 2, we can show that

$$\max_{P\in\Gamma(\tilde{P})} \mathbb{E}[L(Y,\hat{\psi}_n(\mathbf{X}))] = \min_{\mathbf{A}} \mathbb{E}_{\tilde{P}_X}\left[ \sup_{P_{Y|\mathbf{x}}\in\mathcal{P}_{\mathcal{Y}}} \mathbb{E}\left[L(Y,\hat{\psi}_n(\mathbf{X}))|\mathbf{X}=\mathbf{x}\right] + \mathbb{E}[\boldsymbol{\theta}(Y)|\mathbf{X}=\mathbf{x}]^T\mathbf{A}\mathbf{X} \right]$$

$$- \mathbb{E}_{\tilde{P}}[\boldsymbol{\theta}(Y)^T\mathbf{A}\mathbf{X}] + \sum_{i=1}^{t} \epsilon_i\|\mathbf{A}_i\|_*$$

$$\le \mathbb{E}_{\tilde{P}_X}\left[ \sup_{P_{Y|\mathbf{X}=\mathbf{x}}\in\mathcal{P}_{\mathcal{Y}}} \mathbb{E}\left[L(Y,\hat{\psi}_n(\mathbf{X}))|\mathbf{X}=\mathbf{x}\right] + \mathbb{E}[\boldsymbol{\theta}(Y)|\mathbf{X}]^T\hat{\mathbf{A}}_n\mathbf{X} \right]$$

$$- \mathbb{E}_{\tilde{P}}[\boldsymbol{\theta}(Y)^T\hat{\mathbf{A}}_n\mathbf{X}] + + \sum_{i=1}^{t} \epsilon_i\|\hat{\mathbf{A}}_{n_i}\|_*$$

$$= \mathbb{E}_{\tilde{P}}\left[ F_{\boldsymbol{\theta}}(\hat{\mathbf{A}}_n\mathbf{X}) - \boldsymbol{\theta}(Y)^T\hat{\mathbf{A}}_n\mathbf{X} \right] + \sum_{i=1}^{t} \epsilon_i\|\hat{\mathbf{A}}_{n_i}\|_*.$$

Here we first upper bound the minimum by taking the specific $\mathbf{A} = \hat{\mathbf{A}}_n$. Then the equality holds because $\hat{\psi}_n$ is a robust Bayes decision rule against $\Gamma(\hat{P}_n)$ and therefore adding the second term based on $\hat{\mathbf{A}}_n\mathbf{x}$, $\hat{\psi}_n(\mathbf{x})$ results in a saddle point for the following problem

$$F_{\boldsymbol{\theta}}(\hat{\mathbf{A}}_n\mathbf{X}) = \sup_{P\in\mathcal{P}_{\mathcal{Y}}} H(Y) + \mathbb{E}[\boldsymbol{\theta}(Y)]^T\hat{\mathbf{A}}_n\mathbf{X}$$

$$= \sup_{P\in\mathcal{P}_{\mathcal{Y}}} \inf_{\zeta\in\mathcal{Z}} \mathbb{E}[L(Y,\zeta)] + \mathbb{E}[\boldsymbol{\theta}(Y)]^T\hat{\mathbf{A}}_n\mathbf{X}$$

$$= \sup_{P\in\mathcal{P}_{\mathcal{Y}}} \mathbb{E}[L(Y,\hat{\psi}_n(\mathbf{X}))] + \mathbb{E}[\boldsymbol{\theta}(Y)]^T\hat{\mathbf{A}}_n\mathbf{X}.$$

Therefore, by Theorem 2 we have

$$\max_{P\in\Gamma(\tilde{P})} \mathbb{E}[L(Y,\hat{\psi}_n(\mathbf{X}))] - \max_{P\in\Gamma(\tilde{P})} \mathbb{E}[L(Y,\tilde{\psi}(\mathbf{X}))] \le \tag{28}$$

$$\mathbb{E}_{\tilde{P}}\left[F_{\boldsymbol{\theta}}(\hat{\mathbf{A}}_n\mathbf{X}) - \boldsymbol{\theta}(Y)^T\hat{\mathbf{A}}_n\mathbf{X}\right] + \sum_{i=1}^{t} \epsilon_i\|\hat{\mathbf{A}}_{n_i}\|_* - \mathbb{E}_{\tilde{P}}\left[F_{\boldsymbol{\theta}}(\tilde{\mathbf{A}}\mathbf{X}) - \boldsymbol{\theta}(Y)^T\tilde{\mathbf{A}}\mathbf{X}\right] - \sum_{i=1}^{t} \epsilon_i\|\tilde{\mathbf{A}}_i\|_*.$$

As a result, we only need to bound the uniform convergence rate in the other side of the duality. Note that by the definition of $F_{\boldsymbol{\theta}}$,

$$\forall\, P\in\mathcal{P}_{\mathcal{Y}},\ \mathbf{z}\in\mathbb{R}^t:\quad F_{\boldsymbol{\theta}}(\mathbf{z}) - \mathbb{E}_P[\boldsymbol{\theta}(Y)]^T\mathbf{z} \ge H_P(Y) \ge 0. \tag{29}$$

Hence, $\forall\,\mathbf{A}:\ F_{\boldsymbol{\theta}}(\mathbf{AX})-\mathbb{E}[\boldsymbol{\theta}(Y)]^T\mathbf{AX}\geq 0$ and comparing the optimal solution to the RHS of the duality equation in Theorem 2 to the case $\mathbf{A}=\mathbf{0}$ implies that for any possible solution $\mathbf{A}^*$

$$\forall\,1\leq i\leq t:\quad \epsilon_i\|\mathbf{A}_i^*\|_q\leq\sum_{j=1}^t\epsilon_j\|\mathbf{A}_j^*\|_q\leq F_{\boldsymbol{\theta}}(\mathbf{0})=\max_{P\in\mathcal{P}_{\mathcal{Y}}}H(Y)=M. \tag{30}$$

Hence, since $1\leq q\leq 2$, we only need to bound the uniform convergence rate in a bounded space where $\forall\,1\leq i\leq t:\|\mathbf{A}_i\|_2\leq\|\mathbf{A}_i\|_q\leq\frac{M}{\epsilon_i}$. Also, applying the derivative property of the conjugate relationship indicates that $\partial F_{\boldsymbol{\theta}}(\mathbf{z})$ is a subset of the convex hull of $\{\mathbb{E}[\boldsymbol{\theta}(Y)]:\ P\in\mathcal{P}_{\mathcal{Y}}\}$. Therefore, when $\theta(Y)$ includes only one variable, for any $u\in\partial F_{\boldsymbol{\theta}}(z)$ we have $|u|\leq L$, and $F_{\boldsymbol{\theta}}(z)-\theta(Y)z$ is $2L$-Lipschitz in $z$. As a result, since $\|\mathbf{X}\|_2\leq B$ and $|\theta(Y)|\leq L$ for any $\boldsymbol{\alpha}_1,\boldsymbol{\alpha}_2\in\mathbb{R}^d$ such that $\|\boldsymbol{\alpha}_i\|_2\leq\frac{M}{\epsilon}$,

$$\forall\,\mathbf{x}_1,\mathbf{x}_2,y_1,y_2:\ \big[\,F_{\boldsymbol{\theta}}(\boldsymbol{\alpha}_1^T\mathbf{x}_1)-\theta(y_1)\boldsymbol{\alpha}_1^T\mathbf{x}_1\,\big]-\big[\,F_{\boldsymbol{\theta}}(\boldsymbol{\alpha}_2^T\mathbf{x}_2)-\theta(y_2)\boldsymbol{\alpha}_2^T\mathbf{x}_2\,\big]\leq\frac{4BML}{\epsilon} \tag{31}$$

Consequently, we can apply standard uniform convergence results given convexity-Lipschitzness-boundedness [3] to show that for any $\delta>0$ with a probability at least $1-\delta$

$$\forall\,\boldsymbol{\alpha}\in\mathbb{R}^d,\|\boldsymbol{\alpha}\|_2\leq\frac{M}{\epsilon}: \tag{32}$$

$$\mathbb{E}_{\tilde{P}}\big[F_{\boldsymbol{\theta}}(\boldsymbol{\alpha}^T\mathbf{X})-\theta(Y)\boldsymbol{\alpha}^T\mathbf{X}\big]-\mathbb{E}_{\hat{P}_n}\big[F_{\boldsymbol{\theta}}(\boldsymbol{\alpha}^T\mathbf{X})-\theta(Y)\boldsymbol{\alpha}^T\mathbf{X}\big]\leq\frac{4BLM}{\epsilon\sqrt{n}}\left(1+\sqrt{\frac{\log(2/\delta)}{2}}\right).$$

Therefore, considering $\hat{\boldsymbol{\alpha}}_n$ and $\tilde{\boldsymbol{\alpha}}$ as the solution to the dual problems corresponding to the empirical and underlying cases, for any $\delta>0$ with a probability at least $1-\delta/2$

$$\mathbb{E}_{\tilde{P}}\big[F_{\boldsymbol{\theta}}(\hat{\boldsymbol{\alpha}}_n^T\mathbf{X})-\theta(Y)\hat{\boldsymbol{\alpha}}_n^T\mathbf{X}\big]+\epsilon\|\hat{\boldsymbol{\alpha}}_n\|_q \tag{33}$$

$$-\mathbb{E}_{\hat{P}_n}\big[F_{\boldsymbol{\theta}}(\hat{\boldsymbol{\alpha}}_n^T\mathbf{X})-\theta(Y)\hat{\boldsymbol{\alpha}}_n^T\mathbf{X}\big]-\epsilon\|\hat{\boldsymbol{\alpha}}_n\|_q\leq\frac{4BLM}{\epsilon\sqrt{n}}\left(1+\sqrt{\frac{\log(4/\delta)}{2}}\right).$$

Since $\hat{\boldsymbol{\alpha}}_n$ is minimizing the objective for $Q=\hat{P}_n$,

$$\mathbb{E}_{\hat{P}_n}\big[F_{\boldsymbol{\theta}}(\hat{\boldsymbol{\alpha}}_n^T\mathbf{X})-\theta(Y)\hat{\boldsymbol{\alpha}}_n^T\mathbf{X}\big]+\epsilon\|\hat{\boldsymbol{\alpha}}_n\|_q \tag{34}$$

$$-\mathbb{E}_{\hat{P}_n}\big[F_{\boldsymbol{\theta}}(\tilde{\boldsymbol{\alpha}}^T\mathbf{X})-\theta(Y)\tilde{\boldsymbol{\alpha}}^T\mathbf{X}\big]-\epsilon\|\tilde{\boldsymbol{\alpha}}\|_q\leq 0.$$

Also, since $\tilde{\boldsymbol{\alpha}}$ does not depend on the samples, the Hoeffding's inequality implies that with a probability at least $1-\delta/2$

$$\mathbb{E}_{\hat{P}_n}\big[F_{\boldsymbol{\theta}}(\tilde{\boldsymbol{\alpha}}^T\mathbf{X})-\theta(Y)\tilde{\boldsymbol{\alpha}}^T\mathbf{X}\big]+\epsilon\|\tilde{\boldsymbol{\alpha}}\|_q \tag{35}$$

$$-\mathbb{E}_{\tilde{P}}\big[F_{\boldsymbol{\theta}}(\tilde{\boldsymbol{\alpha}}^T\mathbf{X})-\theta(Y)\tilde{\boldsymbol{\alpha}}^T\mathbf{X}\big]-\epsilon\|\tilde{\boldsymbol{\alpha}}\|_q\leq\frac{2BML}{\epsilon}\sqrt{\frac{\log(4/\delta)}{2n}}.$$

Applying the union bound, combining (33), (34), (35) shows that with a probability at least $1-\delta$, we have

$$\mathbb{E}_{\hat{P}_n}\big[F_{\boldsymbol{\theta}}(\hat{\boldsymbol{\alpha}}_n^T\mathbf{X})-\theta(Y)\hat{\boldsymbol{\alpha}}_n^T\mathbf{X}\big]+\epsilon\|\hat{\boldsymbol{\alpha}}_n\|_q \tag{36}$$

$$-\mathbb{E}_{\tilde{P}}\big[F_{\boldsymbol{\theta}}(\tilde{\boldsymbol{\alpha}}^T\mathbf{X})-\theta(Y)\tilde{\boldsymbol{\alpha}}^T\mathbf{X}\big]-\epsilon\|\tilde{\boldsymbol{\alpha}}\|_q\leq\frac{4BLM}{\epsilon\sqrt{n}}\left(1+\frac{3}{2}\sqrt{\frac{\log(4/\delta)}{2}}\right).$$

Given (28) and (36), the proof is complete.

Note that we can improve the result in the case $q=1$ by using the same proof and plugging in the Rademacher complexity of the $\ell_1$-bounded linear functions. Here, we replace the assumption that $\|\mathbf{X}\|_2\leq B$ with $\|\mathbf{X}\|_\infty\leq B$ which can be much weaker for high-dimensional $\mathbf{X}$'s.

**Theorem.** *Consider a loss function $L$ with the entropy $H$ and suppose $\boldsymbol{\theta}(Y)$ includes only one element. Let $M=\max_{P\in\mathcal{P}_{\mathcal{Y}}}H(Y)$ be the maximum entropy value over $\mathcal{P}_{\mathcal{Y}}$. Also, take $\|\cdot\|/\|\cdot\|_*$ to be the $\ell_\infty/\ell_1$ pair. Given that $\mathbf{X}$ is a $d$-dimensional vector with $\|\mathbf{X}\|_\infty\leq B$, and $|\theta(Y)|\leq L$, for any $\delta>0$ with probability at least $1-\delta$*

$$\max_{P\in\Gamma(\tilde{P})}\mathbb{E}[L(Y,\hat{\psi}_n(\mathbf{X}))]-\max_{P\in\Gamma(\tilde{P})}\mathbb{E}[L(Y,\tilde{\psi}(\mathbf{X}))]\leq\frac{4BLM}{\epsilon\sqrt{n}}\left(\sqrt{2\log(2d)}+\sqrt{\frac{9\log(4/\delta)}{8}}\right). \tag{37}$$

# 4 0-1 Loss: minimax SVM

## 4.a $F_{\boldsymbol{\theta}}$ derivation

Given the defined one-hot encoding $\boldsymbol{\theta}$ we define $\tilde{\mathbf{z}} = (\mathbf{z}, 0)$ and represent each randomized decision rule $\zeta$ with its corresponding loss vector $\mathbf{L} \in \mathbb{R}^{t+1}$ such that $L_i = L_{\text{0-1}}(i, \zeta)$ denotes the 0-1 loss suffered by $\zeta$ when $Y = i$. It can be seen that $\mathbf{L}$ is a feasible loss vector if and only if $\forall\, i : 0 \leq L_i \leq 1$ and $\sum_{i=1}^{t+1} L_i = t$. Then,

$$F_{\boldsymbol{\theta}}(\mathbf{z}) = \max_{\substack{\mathbf{p} \in \mathbb{R}^{t+1}:\, \mathbf{1}^T\mathbf{p}=1, \\ \forall i:\, 0 \leq p_i}} \quad \min_{\substack{\mathbf{L} \in \mathbb{R}^{t+1}:\, \mathbf{1}^T\mathbf{L}=t, \\ \forall i:\, 0 \leq L_i \leq 1}} \sum_{i=1}^{t+1} p_i(\tilde{z}_i + L_i). \tag{38}$$

Hence, Sion's minimax theorem implies that the above minimax problem has a saddle point. Thus,

$$F_{\boldsymbol{\theta}}(\mathbf{z}) = \min_{\substack{\mathbf{L} \in \mathbb{R}^{t+1}:\, \mathbf{1}^T\mathbf{L}=t, \\ \forall i:\, 0 \leq L_i \leq 1}} \max_{1 \leq i \leq t+1} \{\tilde{z}_i + L_i\}. \tag{39}$$

Consider $\sigma$ as the permutation sorting $\tilde{\mathbf{z}}$ in a descending order and for simplicity let $\tilde{z}_{(i)} = \tilde{z}_{\sigma(i)}$. Then,

$$\forall 1 \leq k \leq t+1: \quad \max_{1 \leq i \leq t+1} \{\tilde{z}_i + L_i\} \geq \frac{1}{k} \sum_{i=1}^{k} [\tilde{z}_{\sigma(i)} + L_{\sigma(i)}] \geq \frac{k - 1 + \sum_{i=1}^{k} \tilde{z}_{(i)}}{k}, \tag{40}$$

which is independent of the value of $L_i$'s. Therefore,

$$\max_{1 \leq k \leq t+1} \frac{k - 1 + \sum_{i=1}^{k} \tilde{z}_{(i)}}{k} \leq F_{\boldsymbol{\theta}}(\mathbf{z}). \tag{41}$$

On the other hand, if we let $k_{\max}$ be the largest index satisfying $\sum_{i=1}^{k_{\max}} [\tilde{z}_{(i)} - \tilde{z}_{(k_{\max})}] < 1$ and define

$$\forall\, 1 \leq j \leq t+1: \quad L^*_{\sigma(j)} = \begin{cases} \dfrac{k_{\max} - 1 + \sum_{i=1}^{k_{\max}} \tilde{z}_{(i)}}{k_{\max}} - \tilde{z}_{(j)} & \text{if } \sigma(j) \leq k_{\max} \\ 1 & \text{if } \sigma(j) > k_{\max}, \end{cases} \tag{42}$$

we can simply check that $\mathbf{L}^*$ is a feasible point since $\sum_{i=1}^{t+1} L^*_i = t$ and $L^*_{\sigma(k_{\max})} \leq 1$ so for all $i$'s $L^*_{\sigma(i)} \leq 1$. Also, $L^*_{\sigma(1)} \geq 0$ because $\tilde{z}_{(1)} - \tilde{z}_{(j)} < 1$ for any $j \leq k_{\max}$, so for all $i$'s $L^*_{\sigma(i)} \geq 0$. Then for this $\mathbf{L}^*$ we have

$$F_{\boldsymbol{\theta}}(\mathbf{z}) \leq \max_{1 \leq i \leq t+1} \{\tilde{z}_i + L^*_i\} = \frac{k_{\max} - 1 + \sum_{i=1}^{k_{\max}} \tilde{z}_{(i)}}{k_{\max}}. \tag{43}$$

Therefore, (41) holds with equality and achieves its maximum at $k = k_{\max}$,

$$F_{\boldsymbol{\theta}}(\mathbf{z}) = \max_{1 \leq k \leq t+1} \frac{k - 1 + \sum_{i=1}^{k} \tilde{z}_{(i)}}{k} = \frac{k_{\max} - 1 + \sum_{i=1}^{k_{\max}} \tilde{z}_{(i)}}{k_{\max}}. \tag{44}$$

Moreover, $\mathbf{L}^*$ corresponds to a randomized robust Bayes act, where we select label $i$ according to the probability vector $\mathbf{p}^* = \mathbf{1} - \mathbf{L}^*$ that is

$$\forall 1 \leq j \leq t+1: \quad p^*_{\sigma(j)} = \begin{cases} \dfrac{1 - \sum_{i=1}^{k_{\max}} \tilde{z}_{(i)}}{k_{\max}} + \tilde{z}_{(j)} & \text{if } \sigma(j) \leq k_{\max} \\ 0 & \text{if } \sigma(j) > k_{\max}. \end{cases} \tag{45}$$

Given $F_{\boldsymbol{\theta}}$ we can simply derive the gradient $\nabla F_{\boldsymbol{\theta}}$ to find the entropy maximizing distribution. Here if the inequality $\sum_{i=1}^{k_{\max}} [\tilde{\mathbf{z}}_{\sigma(i)} - \tilde{\mathbf{z}}_{(k_{\max}+1)}] \geq 1$ holds strictly, which is true almost everywhere on $\mathbb{R}^t$,

$$\forall 1 \leq i \leq t: \quad \left(\nabla F_{\boldsymbol{\theta}}(\mathbf{z})\right)_i = \begin{cases} 1/k_{\max} & \text{if } \sigma(i) \leq k_{\max}, \\ 0 & \text{Otherwise.} \end{cases} \tag{46}$$

If the inequality does not strictly hold, $F_{\boldsymbol{\theta}}$ is not differentiable at $\mathbf{z}$; however, the above vector still lies in the subgradient $\partial F_{\boldsymbol{\theta}}(\mathbf{z})$.

## 4.b    Sufficient Conditions for Applying Theorem 1.a

As supposed in Theorem 1.a, the space $\mathcal{X}$ should be finite in order to apply that result. Here, we show for the proposed structure on $\Gamma(Q)$ one can relax this condition while Theorem 1.a still holds. It is because, as shown in the proofs of Theorems 2 and 3, we have

$$
\inf_{\psi \in \Psi} \max_{P \in \Gamma(\tilde{P})} \mathbb{E}[L(Y, \psi(\mathbf{X}))] = \inf_{\psi \in \Psi} \min_{\mathbf{A}} \mathbb{E}_{\tilde{P}_X} \left[ \sup_{P_{Y|\mathbf{X}} \in \mathcal{P}_{\mathcal{Y}}} \mathbb{E}[L(Y, \psi(\mathbf{X}))|\mathbf{X} = \mathbf{x}] \right.
$$
$$
\left. + \mathbb{E}[\boldsymbol{\theta}(Y)|\mathbf{X} = \mathbf{x}]^T \mathbf{A}\mathbf{X} \right] - \mathbb{E}_{\tilde{P}}[\boldsymbol{\theta}(Y)^T \mathbf{A}\mathbf{X}] + \sum_{i=1}^{t} \epsilon_i \|\mathbf{A}_i\|_*
$$
$$
= \min_{\mathbf{A}} \mathbb{E}_{\tilde{P}_X} \left[ \inf_{\psi(\mathbf{x}) \in \mathcal{Z}} \sup_{P_{Y|\mathbf{X}} \in \mathcal{P}_{\mathcal{Y}}} \mathbb{E}[L(Y, \psi(\mathbf{x}))|\mathbf{X} = \mathbf{x}] \right.
$$
$$
\left. + \mathbb{E}[\boldsymbol{\theta}(Y)|\mathbf{X} = \mathbf{x}]^T \mathbf{A}\mathbf{X} \right] - \mathbb{E}_{\tilde{P}}[\boldsymbol{\theta}(Y)^T \mathbf{A}\mathbf{X}] + \sum_{i=1}^{t} \epsilon_i \|\mathbf{A}_i\|_*.
$$

Therefore, given this structure the minimax problem decouples across different $\mathbf{x}$'s. Hence, the assumption of finite $\mathcal{X}$ is no longer needed, because as long as $\boldsymbol{\theta}$ is a bounded function (which is true for the one-hot encoding $\boldsymbol{\theta}$), the rest of assumptions suffice to guarantee the existence of a saddle point given $\mathbf{X} = \mathbf{x}$ for any $\mathbf{x}$.

# 5    Quadratic Loss: Linear Regression

## 5.a    $F_{\boldsymbol{\theta}}$ derivation

Here, we find $F_{\boldsymbol{\theta}}(\mathbf{z}) = \max_{P \in \mathcal{P}_{\mathcal{Y}}} H(Y) + \mathbb{E}[\boldsymbol{\theta}(Y)]^T \mathbf{z}$ for $\boldsymbol{\theta}(Y) = Y$ and $\mathcal{P}_{\mathcal{Y}} = \{P_Y : \mathbb{E}[Y^2] \leq \rho^2\}$. Since for quadratic loss $H(Y) = \text{Var}(Y) = \mathbb{E}[Y^2] - \mathbb{E}[Y]^2$, the problem is equivalent to

$$
F_{\boldsymbol{\theta}}(z) = \max_{\mathbb{E}[Y^2] \leq \rho^2} \mathbb{E}[Y^2] - \mathbb{E}[Y]^2 + z\mathbb{E}[Y] \tag{47}
$$

As $\mathbb{E}[Y]^2 \leq \mathbb{E}[Y^2]$, it can be seen for the solution $\mathbb{E}_{P^*}[Y^2] = \rho^2$ and therefore we equivalently solve

$$
F_{\boldsymbol{\theta}}(z) = \max_{|\mathbb{E}[Y]| \leq \rho} \rho^2 - \mathbb{E}[Y]^2 + z\mathbb{E}[Y] = \begin{cases} \rho^2 + z^2/4 & \text{if } |z/2| \leq \rho \\ \rho|z| & \text{if } |z/2| > \rho. \end{cases} \tag{48}
$$

## 5.b    Applying Theorem 2 while restricting $\mathcal{P}_{\mathcal{Y}}$

For the quadratic loss, we first change $\mathcal{P}_{\mathcal{Y}} = \{P_Y : \mathbb{E}[Y^2] \leq \rho^2\}$ and then apply Theorem 2. Note that by modifying $F_{\boldsymbol{\theta}}$ based on the new $\mathcal{P}_{\mathcal{Y}}$ we also solve a modified version of the maximum conditional entropy problem

$$
\max_{\substack{P: P_{\mathbf{X},Y} \in \Gamma(Q) \\ \forall \mathbf{x}: P_{Y|\mathbf{X}=\mathbf{x}} \in \mathcal{P}_{\mathcal{Y}}}} H(Y|\mathbf{X}) \tag{49}
$$

In the case $\mathcal{P}_{\mathcal{Y}} = \{P_Y : \mathbb{E}[Y^2] \leq \rho^2\}$ Theorem 2 remains valid given the above modification in the maximum conditional entropy problem. This is because the inequality constraint $\mathbb{E}[Y^2|\mathbf{X} = \mathbf{x}] \leq \rho^2$ is linear in $P_{Y|\mathbf{X}=\mathbf{x}}$, and thus the problem is still convex and strong duality holds as well. Also, when we move the constraints of $\mathbf{w}_i = \mathbb{E}_P[\theta_i(Y)\mathbf{X}]$ to the objective function, we get a similar dual problem

$$
\min_{\mathbf{A}} \sup_{\substack{P_{Y|\mathbf{X}}, \mathbf{w}: \\ \forall \mathbf{x}: P_{Y|\mathbf{X}=\mathbf{x}} \in \mathcal{P}_{\mathcal{Y}}}} \mathbb{E}_{Q_{\mathbf{x}}} \left[ H_P(Y|\mathbf{X} = \mathbf{x}) + \sum_{i=1}^{t} \mathbb{E}[\theta_i(Y)|\mathbf{X} = \mathbf{x}]\mathbf{A}_i\mathbf{X} \right] + \sum_{i=1}^{t} [I_{C_i}(\mathbf{w}_i) - \mathbf{A}_i\mathbf{w}_i]
$$

(50)

Following the next steps of the proof of Theorem 2, we complete the proof assuming the modification on $F_{\boldsymbol{\theta}}$ and the maximum conditional entropy problem.

### 5.c   Derivation of group lasso

To derive the group lasso problem, we slightly change the structure of $\Gamma(Q)$. First assume the subsets $I_1, \ldots, I_k$ are disjoint. Consider a set of distributions $\Gamma_{\mathrm{GL}}(Q)$ with the following structure

$$
\Gamma_{\mathrm{GL}}(Q) = \{\, P_{\mathbf{X},Y} : P_{\mathbf{X}} = Q_{\mathbf{X}} \,, \tag{51}
$$
$$
\forall\, 1 \leq j \leq k: \ \ \| \mathbb{E}_P \left[ Y\mathbf{X}_{I_j} \right] - \mathbb{E}_Q \left[ Y\mathbf{X}_{I_j} \right] \| \leq \epsilon_j \,\}.
$$

Now we prove a modified version of Theorem 2,

$$
\max_{P \in \Gamma_{\mathrm{GL}}(Q)} H(Y|\mathbf{X}) \ = \ \min_{\boldsymbol{\alpha}} \ \mathbb{E}_Q \left[ F_{\boldsymbol{\theta}}(\boldsymbol{\alpha}^T \mathbf{X}) - Y\boldsymbol{\alpha}^T \mathbf{X} \right] + \sum_{j=1}^{k} \epsilon_j \|\boldsymbol{\alpha}_{I_j}\|_*. \tag{52}
$$

To prove this identity, we can use the same proof provided for Theorem 2. We only need to redefine $\tilde{\mathbf{E}}_j = \mathbb{E}_Q \left[ Y\mathbf{X}_{I_j} \right]$ and $C_j = \{\mathbf{u} : \|\mathbf{u} - \tilde{\mathbf{E}}_j\| \leq \epsilon_j\}$ for $1 \leq j \leq k$. Notice that here $t = 1$. Using the same technique in that proof, the dual problem can be formulated as

$$
\min_{\boldsymbol{\alpha}} \ \sup_{P_{Y|\mathbf{X}},\mathbf{w}} \ \mathbb{E}_{Q_{\mathbf{x}}} \left[ H_P(Y|\mathbf{X}=\mathbf{x}) + \mathbb{E}[Y|\mathbf{X}=\mathbf{x}]\boldsymbol{\alpha}^T \mathbf{X} \right] + \sum_{j=1}^{k} \left[ I_{C_j}(\mathbf{w}_{I_j}) - \boldsymbol{\alpha}_{I_j}\mathbf{w}_{I_j} \right]. \tag{53}
$$

Similarly, we can decouple and simplify the above problem to derive the RHS of (52). Then, if we let $\|\cdot\|$ be the $\ell_q$-norm, we will get the group lasso problem with the $\ell_{1,p}$ regularizer.

If the subsets are not disjoint, we can create new copies of each feature corresponding to a repeated index, such that there will be no repeated indices after adding the new features. Note that since $P_{\mathbf{X}}$ has been fixed over $\Gamma_{\mathrm{GL}}(Q)$ adding the extra copies of original features does not change the maximum-conditional entropy problem. Hence, we can use the result proven for the disjoint case and derive the overlapping group lasso problem.

## Footnotes

[1] $L(y, \zeta)$ is a short-form for $E[L(y, A)]$ where $A \in \mathcal{A}$ is a random action distributed according to $\zeta$.