[Reviews · NeurIPS 2016]

Reviewer 1

Summary

The paper proposes the principle of maximum conditional entropy as a principled way for minimax supervised learning. Under some "structure", they reduce it to learning generalized linear models. They develop minimax SVM through this approach, which outperforms standard SVM.

Qualitative Assessment

The principle of maximum conditional entropy is an interesting approach to supervised learning that is well motivated. Comments: 1. What is the difference between principle of maximum entropy and principle of maximum conditional entropy? Note that H(Y|X) = H(Y,X) - H(X). In the analysis, tau(Q) contains all distributions that contain same marginal as P_X. Hence, all the distributions in tau(Q) have the same H(X). Thus the proposed approach is same as maximizing H(Y,X) (under some constraints) which is nothing but actual entropy as apposed to conditional entropy. In this sense, how is the proposed approach different from principle of maximum entropy? 2. The current set of results do not include any theoretical guarantees on the statistical complexities of learning under this approach. For example, one of the interesting properties of ERM is that it characterizes the statistical complexity of the learning problem in terms of VC dimension / Radamacher complexity. The results would be particularly interesting, if there is a similar theory behind conditional entropy. 3. The choice of tau(hat(p)), where the authors restrict the model to the set of distributions to be the ones which have the same marginal seems less motivated. It would be good to add more explanation.

Confidence in this Review

2-Confident (read it all; understood it all reasonably well)


Reviewer 2

Summary

In this article, the authors propose to optimize of the conditional entropy in order to build algorithm in high dimension settings. First, they introduce the notations and give some examples. Then, they show that the maximization boils down to regularized maximum likelihood problem and give a bound for the excess risk. A small numerical illustration is given at the end.

Qualitative Assessment

Overall, I had a really hard time to read the paper. To this point, I still do not know if the feature space \cal{X} is discrete or continuous or whatever. I also had a hard time with the function \theta, because the examples are two pages after the first utilization. I believe that the main interest of this paper is to give a better understanding of the classical algorithm or at least another perspective. However, the quality of the writing is not good enough to make it clear.

Confidence in this Review

2-Confident (read it all; understood it all reasonably well)


Reviewer 3

Summary

This paper develops a notion of maximum conditional entropy for arriving at robust Bayes decision rules, where the entropies are generalizations of the usual Shannon entropy, beyond the case of log loss. The formulation is similar to that of Grunwald and Dawid but goes farther (and becomes more relevant for standard learning problems) by allowing for input and label random variables and hence conditional distributions. The first main result shows that (under regularity conditions) the Bayes rule for the distribution that maximizes conditional entropy is also a robust Bayes decision rule (against a set of distributions $\Gamma$). The authors then show an interesting duality result that re-expresses the problem of maximizing conditional entropy as a certain regularized minimization problem that, for exponential families and log loss, corresponds to regularized maximum likelihood. They also connect the suggested procedure to an excess risk bound, showing that with high probability, using their procedure for an empirical distribution guarantees low excess risk versus the hypothesis one obtains when using their procedure (which is still regularized) with oracle access to the true distribution; this result apparently is only for log loss, as per the leadup to Theorem 3. Finally, they show important derivations of key quantities for the method in the case of logistic, 0-1, and quadratic loss, corresponding previously known learning methods or new ones (in the case of minimax SVM). Initial experiments are promising.

Qualitative Assessment

\documentclass{article} \begin{document} This work represents a very interesting and original generalization of the work of Gr\"unwald and Dawid to the case of learning with inputs $x$. The technical results appear to be correct and the experimental results (which I think are quite preliminary) suggest the minimax SVM might be a good idea. I think the idea of robust Bayes decision rules makes sense and the authors show how under squared loss a connection to the Huber loss emerges. My main comment is that the paper itself is a somewhat difficult read due to terseness at key places, which might limit the impact of the paper. So, the rest of my comments are just geared towards improving the clarity of the paper. Technically, in every instance where the authors apply Danskin's theorem, it was not really clear what form of Danskin's theorem was being used, and therefore it was difficult to follow the derivation. For instance, the proof of Theorem 2 (in the appendix) is clear up until the end ``Furthermore, applying Danskin's theorem to (9) implies...''. You should provide a citation for the form of Danskin's theorem being used, or just reproduce the theorem in the appendix. Using the form that I found, I could not reproduce (12); I think this comes down to the authors skipping quite a few steps. Also, in equation (24) of the appendix, I could not see how Danskin's theorem give (24), but this might be clarified once you mention the form of Daskins's theorem being used. I found it difficult to follow the argument from lines 187-194 of the main text, including Corollary 1. It would be useful to explain this better, especially given that you can fit another paragraph in the paper and still stay within 8 pages. In particular, why does the uniform conditional distribution appear in equation (18)? Perhaps you can explain the ``it can be seen that'' with a few sentences. I think the geometric understanding provided is one of the highlights of the paper, so it is unfortunate if the reader cannot fully grasp what you have in mind. Minor comments: In equation (10) in the appendix, you are missing a minus sign in front of $\mathbf{A}_i$ both in the middle expression and the rightmost expression. This minus sign does appear in the first line of equation (11), so the mistake apparently was just a typo. On line 26 of the appendix, you say that ``problem (5) is convex''. But the objective is concave being the entropy (a minimum of a linear function) and indicator are both concave. \end{document}

Confidence in this Review

2-Confident (read it all; understood it all reasonably well)


Reviewer 4

Summary

The authors consider the problem of corruption/robustness in standard classification problems. Under what types of corruption is it possible to get computationally efficient algorithms with meaningful results is the crux of the problem attacked. The paper shows that under appropriate conditions on the corruption, one can get the "Bayes classifier" in the modified setting setting in a computationally efficient manner.

Qualitative Assessment

Technically the paper is interesting, and significant, as it gives important resultants on optimal classifiers under robust conditions, but the presentation can be improved significantly. My main comments are given below. Firstly, the focus of the paper is robustness and optimality under corruption but the introduction barely touches upon this and talks about high dimensions and sample complexity issues, which are very peripheral issues in the paper. The introduction put me off quite a bit and it took a while to see the real focus of the paper. Secondly, it would be very interesting to see a good commentary on the effect of the class of probabilities $\Gamma(P)$, which I now see as the the set of transformations, the application of one of which gives the true distribution. For instance while it may look that the larger the set $\Gamma$ the stronger the results are, it is not the case. Setting $\Gamma(P)$ to be the set of all distributions would yield a trivial optimal classifier that essentially gives up on predicting. So $\Gamma$ has interesting restrictions on all sides, it must not be too small, or too big, and it must also allow efficient computation. What types of corruption would fall under the family given by Equation 13? Would something like label noise corruption be a good fit here? (the marginals being assumed to be the same is reminiscent of label noise like corruption). Can any other types of corruption be approximated by equation 13? Are there other settings for $\Gamma(P)$ other than in equation 13, where things are interesting? Thirdly, Theorem 3 does not seem to actually fit in with the rest of the robustness ideology. It does not even use the set $\Gamma(P)$ in its results. While it is good to know that using the algorithm with finite samples yields results as good as using the distribution itself, the effect of $\Gamma$ is not visible at all. In effect the only "corruption" being considered here is the one due to finite sample size -- many other interesting corruptions can be considered as mentioned previously. Minor comment: How is equation 19, obtained? In particular, equation 15 requires a loss for the definition of entropy, what loss is being used here? the log-loss?

Confidence in this Review

1-Less confident (might not have understood significant parts)


Reviewer 5

Summary

This paper proposes a new minimax framework of statistical learning and a principle of maximum conditional entropy to help find solution of the minimax problem. This is a general framework which gives logistic regression, linear regression, and some variant of SVM by using different loss functions. The experiment results show the minimax SVM derived from this framework has good performance.

Qualitative Assessment

The paper is well-organized. The proposed minimax framework is nice and general. One minor issue is that I don't see why Theorem 3 gives the upper bound of the gap between two losses, since Theorem 1 in [16] only gives a bound on the gap between *regularized* loss. Besides, the connection between the proposed framework and ERM is not sound enough since the regularizer derived from this framework and the one for Theorem 3 is different. It would be better if the paper could discuss how much the difference is by replacing ||A_i||_* with ||A_i||^2_*.

Confidence in this Review

1-Less confident (might not have understood significant parts)


Reviewer 6

Summary

The paper leverages a minimax approach to create a new kind of binary classifier, the minimax SVM. This was motived by the complexity of performing good probabilities estimate otherwise and generalization on the REM method. In the end, the framework is used to develop some decision rules for different binary classification losses and illustrated through some simple standard classification tasks. The resulting classifier can reach better accuracy than standard SVM for some of the datasets.

Qualitative Assessment

The approach is interesting by showing how minimax can be used for supervised learning. The results are promising and the model derivation is well motivated. It is however lacking some discussions on the extensions to multi-label classification and the numerical results concerning computation time to fit the model compared to SVM. The classification result table is not easily readable with the bold fonts oppose to the standard fonts. Finally, some extra motivations to use a minimax approach could be usefull in addition of the presented reason which can be tackle by standard PGM. Also overall paragraphs are sometimes hard to follow due to the succession of definitions and results without many comments and insights on what they represent.

Confidence in this Review

2-Confident (read it all; understood it all reasonably well)